# Nutritional Benefits and Consumer Acceptance of Maize Chips Combined with Alternative Flours

**DOI:** 10.3390/foods14050864

**Published:** 2025-03-03

**Authors:** Jesús Rodríguez-Miranda, Meliza Peña, Miriam Rivera, Jason Donovan

**Affiliations:** 1International Maize and Wheat Improvement Center (CIMMYT), Carretera Mexico-Veracruz Km. 45, El Batan 56237, Texcoco, Mexico; miriam.rivera@unah.edu.hn (M.R.); jdonovan@idrc.ca (J.D.); 2Tecnológico Nacional de México/Instituto Tecnológico de Tuxtepec, Calzada Victor Bravo Ahuja, No. 561, Col. Predio el Paraíso, Tuxtepec 68350, Oaxaca, Mexico; 3National Autonomous University of Honduras (UNAH), Campus Comayagua, Intercambio Comayagua, Comayagua 12101, Honduras; 4International Development Research Centre (IDRC), Torre Mapfre, Juncal 1385, Montevideo 11000, Departamento de Montevideo, Uruguay

**Keywords:** alternative flours, maize, nutrition, sensory acceptance, tortilla chips

## Abstract

This study evaluated the nutritional composition, techno-functional properties, and sensory acceptance of tortilla chips made from alternative flours derived from local ingredients, including maize, beet, flaxseed, bean, and chia. Three blends were assessed: maize with beans, maize with beet, and maize with chia–flaxseed. Significant differences (*p* < 0.05) were observed in the flours’ moisture, ash, protein, lipid, and mineral content. Flaxseed flour exhibited the highest protein content (40.03 g/100 g), while chia flour was notable for its lipid (32.25 g/100 g) and fiber (38.51 g/100 g) content. Bean and chia flour were rich in iron and zinc. Sensory evaluations, conducted with 300 consumers in Honduras, revealed general acceptance of all blends, with maize chips enriched with chia–flaxseed showing the highest preference (47.2%). Approximately 50% of participants reported consuming tortilla chips weekly, prioritizing taste, freshness, and price. Notably, over 40% expressed willingness to pay a premium for more nutritious, baked options. These results underscore the potential of alternative flours to enhance local diets and foster healthier eating habits. Moreover, the positive consumer response highlights a significant market opportunity for small and medium-sized enterprises (SMEs), promoting awareness of nutrition and public health in Honduras.

## 1. Introduction

The growing concerns about public health and nutrition have motivated a shift in consumer preferences toward healthier and more functional foods. This trend is particularly relevant in Latin America, where maize, a staple food, has become the focus of food innovations seeking to improve its nutritional profile. According to the Food and Agriculture Organization of the United Nations (FAO), maize is one of the most important crops worldwide, not only for its nutritional value but also for its role in food security [1]. However, traditional maize-based foods often lack certain essential nutrients (e.g., proteins). that could be improved by incorporating alternative flours. Recent research suggests that the use of flaxseed, chia, beet, and bean flours can enrich maize-based foods, increasing their protein, lipid, fiber, and micronutrient content [2,3,4]. Flaxseed and chia are particularly notable for their high omega-3 fatty acid and fiber content, while beetroot is known for its high antioxidant content, including betalains, especially betanin, which provide additional health benefits [3]. Interest in these alternative ingredients is not just a local phenomenon; globally, a trend towards consuming more nutritious and less processed foods has emerged. Ngugi et al. [5] highlighted the trend of consumers being increasingly aware of the relationship between food and health, which influenced their purchasing decisions. This shift has led the food industry to develop innovative products that not only meet dietary needs but also meet quality and taste expectations. 

Previous research has assessed the potential to increase the nutrient content of processed maize foods. Vasisht et al. [6] evaluated the effect of pomegranate peel on the physicochemical and antioxidant properties of tortilla chips prepared from germinated maize and mung bean flour as a healthier snack alternative. They recommend a 5% concentration of peel fried at 180 °C, preserving the sensory attributes and nutrients. Buzgau et al. [7] evaluated the chemical composition and quality characteristics of tortilla chips made from maize flour with chickpea flour and red lentil flour. This study demonstrated that the incorporation of chickpea flour in maize flour chips improved the nutritional qualities of tortilla chips. Zuñiga-Martínez et al. [8] studied the effect of adding avocado paste (AP) to maize chips on their nutritional profile and sensory acceptability. They found that 2% AP was the best concentration to improve the nutritional properties of maize chips without compromising their sensory acceptability.

In the Honduran context, maize is a fundamental component of the diet, consumed primarily as tortillas. However, the market also offers a variety of processed maize products that lack nutritional value. Many of these products are fried and contain preservatives. Despite their cultural and nutritional importance, processed maize foods often present deficiencies in essential nutrients, which can contribute to health problems such as malnutrition and chronic diseases [9].

Incorporating alternative flour in the production of snacks, such as tortilla chips, can offer a viable solution to address these deficiencies while improving the sensory quality of the final product. A crucial aspect of the acceptance of new food products is consumer perception. Sensory evaluation plays a critical role in the acceptance of innovative foods, and sensory characteristics such as taste, texture, and aroma are key determinants in purchasing decisions [10]. Therefore, it is essential to understand how Honduran consumers perceive tortilla chips made with enriched flour and what factors influence their willingness to pay for these products.

This study aimed to evaluate the chemical composition and techno-functional properties of different flours and maize chips made with alternative flour blends. In addition, it sought to analyze the sensory acceptance of these products among Honduran consumers, focusing on their preferences and willingness to pay for more nutritious options. This comprehensive approach will not only contribute to academic research but will also provide valuable information for the food industry and in the development of innovative products that respond to consumer needs and are an option for the development of new products by small and medium-sized enterprises (SMEs).

Through this research, we hope to foster greater awareness of the importance of nutrition and public health in Honduras, while promoting the use of local and nutritious ingredients. This will not only benefit consumers but can also have a positive impact on the local economy by driving innovation among SMEs, enabling them to develop new, healthy and appealing products tailored to evolving consumer preferences.

## 2. Materials and Methods

### 2.1. Raw Materials

In this study, we used nixtamalized maize (*Zea mays* L.) flour from the company MATURAVE S.A de S.V., located at Km 15 of the Olancho highway in Tegucigalpa, Honduras; flaxseeds (*Linum usitatissimum*) marketed by Bosque, Naturamas S. de R.L. of Santiago Puringlas, La Paz, Honduras; chia seeds (*Salvia hispanica*) marketed by Nature’s Earthly Choice, Eagle ID 83616, Idaho, USA); biofortified light red bean (*Phaseolus vulgaris* L.) from Honduras, CIAT/PIF-Zamorano, Honduras; and beet (*Beta vulgaris*) with commercial maturity acquired in the local market of Tegucigalpa, Honduras.

### 2.2. Preparation of Flours

1. Maize flour: The maize was pre-cleaned to remove impurities; then, lime was added to boil it (nixtamalization) for 17 min, and then it was strained, washed, and ground. The ground maize was dehydrated and passed through sieves (until a particle size of approximately 0.5 mm) to remove the pericarp, and then it was packed in sacks.

2. Bean flour: The beans were washed, cleaned of impurities, and boiled in water for 60 min in a pressure cooker (Tramontina S.A. Cutelaria, Av. Raul Giacomoni, 2700, Desvio Machado, Carlos Barbosa, Brasil). Following the method described by Rocha-Guzmán et al. [11], once cooked, the beans were ground with a small amount of the cooking broth until forming a paste, which was then dehydrated in an oven (Ninja, Foodi Air Fry Pro-Horno, Model: FT205CO, Needham, MA 02494, USA) at 45 °C for 5 h. The dried paste was ground using a coffee (KRUPS, Model: F203, Krups North America, Inc. P.O. Box 3900 Peoria, IL 61612, USA) grinder until obtaining a particle size of approximately 0.5 mm.

3. Chia and flaxseed flour: Chia and flaxseed were cleaned manually to remove any impurities. The seeds were then ground in a coffee (KRUPS, Model: F203, Krups North America, Inc., P.O. Box 3900 Peoria, IL 61612, USA) grinder until a particle size of approximately 0.5 mm was obtained.

4. Beet flour: Beets were washed and rinsed thoroughly before being cut into 0.5 cm slices. The slices were dehydrated at 60 °C for 24 h in an oven (Ninja, Foodi Air Fry Pro-Horno, Model: FT205CO, Needham, MA 02494, USA) [12]. The dried slices were then ground in a coffee (KRUPS, Model: F203, USA) grinder to achieve a particle size of approximately 0.5 mm.

### 2.3. Physicochemical Characterization of Raw Materials and Chips

The chemical composition of the raw materials and different chip mixtures was determined according to methods described by AOAC [13]: moisture (Method: 934.01), ash (Method: 942.05), lipids (Method: 954.02), and protein (Method: 2001.11). The insoluble, soluble, and total dietary fiber content of samples was determined using a Total Dietary Fiber Assay Kit (K-TDFR; Megazyme International Ireland Ltd., Bray, Ireland). Carbohydrates were calculated by difference.

Lysine concentration was measured using the colorimetric method described by Galicia-Flores et al. [14] and Palacios-Rojas et al. [15], while tryptophan concentration was assessed with the colorimetric method of Nurit et al. [16].

The mineral concentrations (Al, Cd, Cr, Cu, Fe, Mn, Ti, Zn, Ca, K, P) were determined as described by Palacios-Rojas et al. [15]. The samples (300 mg) were weighed into 100 mL Pyrex tubes. Digestion was initiated by adding 5 mL of a HNO_3_:HClO_4_ mixture (9:1 *v*/*v*). The samples were vortexed, covered with polyethylene wrap, and incubated for pre-digestion overnight at room temperature (25 °C) under a fume hood. Digestion was performed by gradually heating the digestion block from 80 to 225 °C for 4 h. After cooling, 10 mL of HNO_3_ was added, and the sample was mixed. The concentrations of minerals are expressed as mg/kg and were determined via inductively coupled plasma optical emission spectroscopy (ICP-OES; Optima™ 8300 DV, Perkin Elmer, Waltham, MA, USA). The flow rates of plasma gas, nebulizer gas, and auxiliary gas and the peristaltic pump flow rate were fixed at 15.0 L/min, 0.50 L/min, 0.80 L/min, and 1.00 mL/min, respectively.

The color of the samples was determined according to the methodology of Hernández-Santos et al. [12], with a Hunter lab tristimulus colorimeter (MiniScan Hunter Lab, model 45/0 L, Hunter Associates Lab., Ind., Reston, VI, USA). The parameters obtained were *L**, *a*,* and *b**.

In the raw materials, the following were evaluated: For water absorption capacity (WAC) and water solubility capacity (WSA), 1 g of sample was weighed in centrifuge tubes and 10 mL of distilled water was added, shaken for 30 s in a vortex (Vortex-2 Genie, Model G-560, Scientific Industries, Inc., Bohemia, NY USA), and centrifuged for 3500 rpm/15 min (Rotina 380 R, Hettich, Tuttlingen, Germany). The results were expressed as grams of water retained per gram of sample and for WSA as a percentage. For oil absorption capacity (OAC), 1 g of sample was weighed, and 10 mL of maize oil was added and shaken for 30 s in a vortex mixer (Vortex-2 Genie, Model G-560, Scientific Industries, Inc., Bohemia, NY USA), and the mixture was centrifuged at 3500 rpm for 15 min (Rotina 380 R, Hettich, Tuttlingen, Germany). The results were expressed as grams of oil retained per gram of sample [12].

### 2.4. Process of Making Tortilla Chips

The flours of each formulation were mixed manually by adding water until a homogeneous dough with a moisture content of 56 to 57 g/100 g was obtained [17]. The tortillas were then made in a manual tortilla rolling machine (Tortilladora Manual, Model: TM-G, con molde/cortador 5 ½, Marca: Maquinas González, Av. Benito Juárez 310 Ote Centro, Guadalupe Nuevo León 67100 Mexico). The tortillas were cooked on a griddle at 250 ± 10 °C for 30 s on one side and 25 s on the other [18]. After cooking, they were left to rest at room temperature (25 °C) for 15 min before being cut into triangles and baked in an air fryer (Percuitti, Model: AF50-450, Calle Igarsa 13C, 28860, Paracuellos de Jarama, Madrid, Spain) at 180 °C for 15 min on one side and 5 min on the other.

### 2.5. Characterization of Tortillas and Tortilla Chips

1. Degree of contraction (DCo) and degree of volume contraction (DVC): The diameter and thickness of the tortillas and chips were measured before and after cooking using a vernier caliper (Mitutoyo Model: 500-713-20, Mitutoyo Corporation, Sakado 1-Chome, Takatsu-Ku, Kawasaki-shi, Kanagawa 213-8533, Japan) and a digital scale (A&D Weighing, Model: EJ-4100, 1756 Automation Parkway San Jose, CA 95131, USA) [19]. The DCo was calculated according to Equation (1), and the DVC was calculated using Equation (2):(1)$$DCo=\frac{D_{i}-D_{f}}{D_{i}}\times \;100$$(1a)$$D_{Chips}=\frac{D}{4}$$
where *D_i_* = initial diameter and *D_f_* = final diameter, depending on the sample analyzed.(2)$$DVC=\frac{V_{i}-V_{f}}{V_{i}}\times \;100$$(2a)$$V_{T}=\pi r^{2}\times \;h$$(2b)$$V_{Chips}=\frac{\pi r^{2}\times \;h}{4}$$
where *V_i_* = initial volume and *V_f_* = final volume, depending on the sample analyzed (*V_T_* = volume of the tortilla and *V_Chips_* = volume of the chips).

2. Weight loss: Tortillas and chips were weighed before and after cooking, and the percentage of weight loss was calculated using the following formula:(3)$$\%\;Weight=\frac{i_{w}-f_{w}}{i_{w}}\times \;100$$
where *i_w_* = initial weight and *f_w_* = final weight, depending on the sample analyzed.

3. Apparent density (AD): The AD was determined using the seed displacement method [20]. Quinoa seeds were used, which were poured into a 250 mL graduated cylinder to measure the displaced volume. The weight of each sample was determined with a digital scale (A&D Weighing, Model: EJ-4100, 1756 Automation Parkway San Jose, CA 95131, USA). The apparent density was calculated by dividing the weight of the tortillas or tortilla chips by the displaced volume expressed in (kg/m^3^). The test was performed in triplicate.

### 2.6. Selection of Best Formulations

Chips were prepared using different proportions of each blend: maize/bean (MBF), maize/chia/flaxseed (MCF), and maize/beet (MB). Two formulations for each blend were selected according to the preferences of a semi-trained sensory panel, based on the best sensory attributes (flavor, color, appearance, texture and overall acceptance, unpublished data). The proportions of each of the flours and formulations are shown in Table 1 and their appearance is shown in Figure 1.

### 2.7. Consumer Evaluation

The consumer evaluation involved 300 participants, recruited upon entering or exiting supermarkets in the cities of Tegucigalpa, Siguatepeque, and Comayagua in Honduras. A team of interviewers located outside the supermarkets invited shoppers to participate in a tasting session and complete a survey. Each participant tasted two formulations of each chip blend, resulting in 100 completed surveys for each chip type.

The assessment was conducted using a survey developed with the Open Data Kit (ODK) tool. Before data collection, both the survey and the research plan received approval from the CIMMYT Ethics Committee, with approval code IREC 2024.012 (authorized on 21 May 2024). Informed consent was obtained from all participants, ensuring their voluntary participation and anonymity throughout the study. The survey consisted of 22 questions divided into four sections: (1) importance of maize in the diet, (2) sensory evaluation of chips, (3) purchasing willingness scenarios, and (4) sociodemographic profile of respondents.

### 2.8. Statistical Analysis

The results were presented as means with standard deviation. Data analysis was performed using one-way ANOVA and the significance of its variance was verified by Fisher’s test (*p* < 0.05) using Statistica software version 10.

## 3. Results and Discussion

### 3.1. Chemical Composition, Color Parameters, and Techno-Functional Properties of Flour

The chemical and nutritional compositions of the raw materials used are summarized in Table 2. Among the different flours, maize flour had the highest moisture content, while beet flour had the lowest. It is important to emphasize that the moisture content in all samples (ranging from 5.38 to 10.37 g/100 g) is within the recommended limits to avoid deterioration due to the growth of microorganisms. The moisture content in maize flour is lower than the average reported (13.3 g/100 g) in several maize varieties [21]. Beet flour had the highest ash level, while maize flour had the lowest (Table 2). It is noteworthy that flaxseed flour showed high concentrations of trace elements, including aluminum, cadmium, chromium, and titanium. In contrast, bean flour was rich in iron and zinc, essential for immune function and metabolism [22,23], while chia flour stood out for its manganese and potassium content, vital for bone health and muscle function [22]. Beet flour stood out for its high levels of copper, calcium, and phosphorus (Table 2), which contribute to total mineral intake [22]. The high mineral content in these flours suggests possible health benefits.

Chia flour had the highest lipid content, followed by flaxseed flour. In contrast, beet flour had the lowest lipid content (Table 2). Lipids in chia have been reported to be mainly composed of unsaturated fatty acids [24]. Unsaturated fatty acids have shown beneficial health activities [9]. Protein content was highest in flaxseed flour, followed by chia flour and bean flour, supporting their high nutritional value [25,26]. Chia and flaxseed flour exhibited the highest levels of lysine at 1.132 g/100 g and 0.847 g/100 g, respectively, while flaxseed flour contained the highest tryptophan alongside beet flour; both chia and flaxseed flour are rich in lysine and tryptophan, essential amino acids, with flaxseed showing the highest levels of tryptophan [24,26].

In terms of fiber content, bean flour had the highest levels of total fiber and insoluble fiber, closely followed by chia flour (Table 2). Flaxseed flour has the highest concentration of soluble fiber, which helps control cholesterol [27]. The high fiber content found in the flours suggests their potential to formulate functional foods that promote health benefits [3,28,29].

For the color parameters, maize flour had the highest luminosity and a green-tending *a** tone. The rest of the flours had *a** tones with a tendency towards red in different shades (Table 2). In the *b** color parameter, all the flours showed a yellow color tendency with different shades. Beet flour was the flour with the darkest shade. These differences can be attributed to the inherent color characteristics of each type of flour.

Regarding techno-functional properties, chia flour presented the highest WAC and WSC, making it an excellent thickening and gelling agent [30]. This property is attributed to its high fiber content, which contributes to its ability to swell and form gels, thereby improving the texture of food products [30]. Chia flour contributes to a cohesive dough structure in food products, improving the integrity of chips during processing and storage. The gel-forming ability of chia flour can help maintain the structural integrity of chips, preventing them from breaking and ensuring a desirable crunch [3]. Khan et al. [31] mention that the incorporation of chia flour improved sensory attributes such as texture and mouthfeel, making chips more appealing to consumers. Flaxseed flour exhibited the highest OAC, indicating its potential as a functional ingredient in fat-based applications, which can improve the flavor and mouthfeel of various food products [32]. This potential is supported by its rich nutritional profile and functional properties, making it suitable for fat-based applications. The high OAC content improves the sensory attributes of food products [32]. Hussain et al. [33] mention that roasting flaxseed meal improves its water absorption capacity and reduces antinutritional factors, making it more suitable for various food applications, and would favor the preparation of chips.

### 3.2. Chemical Composition and Physical Properties of Chips

#### 3.2.1. Chemical Composition

The moisture content of the different chip formulations varied between 8.17 and 9.84 g/100 g (Table 3). The lowest moisture levels were observed in chips made with maize and beans, which showed significant differences (*p* < 0.05). These values are within the range reported by other authors for bean chips [17]. In contrast, chips made with the maize–beet mixture had the highest moisture content (Table 3), with no significant differences between the two formulations (*p* > 0.05). These differences are in line with the findings of several studies, which highlight the influence of ingredient proportions and baking conditions. The addition of legumes, such as chickpeas and lentils, has been shown to increase moisture content due to their higher water retention properties [7].

Ash content ranged from 1.20 to 5.65 g/100 g, with significant differences (*p* < 0.05) among all formulations. Formulation MBF-1 had the highest ash content, while formulation MCF-2 had the lowest. As anticipated, the formulation with the highest bean concentration had the highest ash content. Studies on *Phaseolus vulgaris* L. suggest that ash content may have a positive impact on nutritional value and health [23,34]. Ash content in the other formulations is within the range reported for maize chips with avocado paste, maize chips, and maize chips with potato [7,35,36]. In terms of mineral composition, the MBF-1 bean formulation presented significantly elevated levels (*p* < 0.05) of copper, iron, zinc, calcium, potassium, and phosphorus.

However, this ash content is lower than that reported by Coorey et al. [3] in chips made from a mixture of maize flour and chia flour (5.89–7.33 g/100 g), and higher than that reported in chips made from a mixture of maize and avocado paste (1.52–1.82 g/100 g) [8].

The beet formulation (MB-1) contained the highest levels (*p* < 0.05) of aluminum and manganese, while the MB-2 formulation presented the highest content of titanium. There were no statistically significant differences (*p* > 0.05) in cadmium content between the chips, with an average concentration of 0.078 mg/kg. The iron and zinc values in the biofortified bean chips are in line with findings emphasizing the importance of these minerals in combating deficiencies, particularly in vulnerable populations [2]. The high levels of aluminum and manganese in the MB-1 formulation suggest its potential for greater nutritional benefits, given that these elements are essential for various biological functions [4].

The lipid content of all baked chips showed significant differences (*p* < 0.05) across all formulations, with the highest lipid content found in the chips with chia and flaxseed and the lowest in the chips with beans (Table 3). The high lipid content in the chia and flaxseed chips is attributed to their rich composition of polyunsaturated fatty acids, which are essential for several bodily functions [37]. All other chips were within the range reported by other authors [8,38], for example, in that reported in chips of mixtures of maize and avocado paste (2.08–3.62 g/100 g) [8] and that reported in corn chips with grape seed extracts (5.3–5.7 g/100 g) [39], and much lower than that reported in chips of mixtures of pomegranate peel, germinated maize, and mung bean flour fried by immersion in oil (21.33–30.00 g/100 g) [6]. Therefore, the low lipid content of the chips developed in this research is evident compared to chips made in the traditional way (fried by immersion in oil).

The protein content was highest in the chips with beans, followed by those made with a blend of maize, chia, and linseed (Table 3). The lowest protein values were found in the maize chips with beet, with significant differences observed (*p* < 0.05). The highest protein content was found in the MBF-1 formulation (Table 3).

The results in the maize and bean formulations coincide with those of Hurtado et al. [40] (13.3–16.6 g/100 g) and Ochoa-Martínez et al. [17] (13.00–13.28 g/100 g), who reported a high protein content in mixtures of these ingredients. The addition of bean flour to maize chips highlights the nutritional benefits of legumes in snacks. Coorey et al. [3] mention that the incorporation of chia flour into chips increases the protein content, and with 5% of chia flour, they maintained excellent sensory qualities and, at the same time, provided important nutritional benefits. On the other hand, Trevisan and Arêas [41] mention that the use of flaxseed flour increased the flaxseed protein content in maize chips.

In terms of lysine content, no significant differences (*p* > 0.05) were observed between MBF-1, MBF-2, MCF-1, and MCF-2 formulations, all averaging 0.480 g/100 g. In contrast, beet formulations presented the lowest lysine values, highlighting the nutritional limitations of beet as a protein source [42]. Similarly, no significant differences (*p* > 0.05) were found in tryptophan content between MBF-1, MBF-2, MCF-2, MB-1, and MB-2 formulations, averaging 0.037 g/100 g; the highest content was found in MCF-1 (Table 3). This highlights the nutritional variability between the different formulations, particularly in essential amino acids.

No significant differences (*p* > 0.05) were observed in total dietary fiber and insoluble dietary fiber content between MB-1 and MB-2 chips. Similarly, soluble dietary fiber content did not show significant differences (*p* > 0.05), averaging 0.055 g/100 g in both types. MBF-1 chips exhibited the highest dietary fiber content at 25.86 g/100 g, closely followed by the MCF-1 formulation. Notably, MBF-1 chips also had the highest insoluble dietary fiber content at 25.09 g/100 g, while MCF-1 contained 19.69 g/100 g. In contrast, chips made with chia and flaxseed had significantly higher soluble fiber content, with MCF-1 at 4.63 g/100 g and MCF-2 at 4.12 g/100 g, indicating a notable difference between these formulations. The soluble fiber content in MCF-1 indicates a greater potential for intestinal health benefits [43]; this content is higher compared to other studies [44].

The fiber content found in chips with maize flour, chia flour, and flaxseed flour is higher than that reported in chips made from a mixture of maize and avocado paste (2.72–5.29 g/100 g) [8], chips made from a mixture of maize flour and chia flour (5.89–7.33 g/100 g) [3], and chips made from a mixture of maize flour and bean flour (11.53–19.21 g/100 g) [38].

In terms of carbohydrate content, significant differences (*p* < 0.05) were observed in all chip formulations, except for those containing beetroot, which did not show statistically significant differences (*p* > 0.05) between them, with an average of 64.98 g/100 g, being the highest value found (Table 3). However, this value is lower than that reported by Agarwal and Chauhan [45] in gluten-free chips made with composite flour (69.5 g/100 g).

#### 3.2.2. Physical Properties

The weight of the chips varied from 2.98 to 3.70 g. No significant differences (*p* > 0.05) were found between the MB-2 and MBF-1 formulations, with an average weight of 3.52 g, the highest value recorded. The lightest chips were those of the MCF-2 formulation with a weight of 2.98 g, which was statistically significant (*p* < 0.05); this can be attributed to its composition, which includes lighter seeds, which affects its overall density [39].

The largest diameter was observed in the MCF-1 and MCF-2 formulations with an average of 29.39 mm, with no significant differences between them (*p* > 0.05), in agreement with the findings that larger particle sizes can improve certain physical properties, such as stability and absorption efficiency [46]. In contrast, chips from the MBF-1 (bean) and MB-1 (beet) formulations had an average diameter of 26.49 mm, with no significant differences between them (*p* > 0.05), reflecting a trend in which variations in raw materials (beans vs. beet) influence diameter size results [47]. Chip thickness ranged from 1.14 to 1.94 mm, with significant differences between all formulations (*p* < 0.05). The greatest thickness was observed in the formulations with beans (Table 3), indicating a possible correlation between the type of ingredient and the physical characteristics.

The chips with the highest shrinkage percentage were the chips made with beans, with values of 32.24% for MBF-1 and 15.94% for MBF-2, showing significant differences (*p* < 0.05) between them. These differences are because the MBF-1 formulation contains a higher concentration of beans (45 g/100 g). Higher concentrations of beans lead to higher shrinkage due to greater moisture loss during baking [48].

In contrast, chips formulated with chia and linseed presented lower shrinkage percentages, with 12.05% for MCF-1 and 10.19% for MCF-2, due to their lower moisture content and, therefore, a lower shrinkage rate, lower shrinkage volume, and lower density [49]. In terms of the degree of volume shrinkage, significant differences (*p* < 0.05) were observed among all the evaluated chips. The highest percentage of volume shrinkage corresponded to the MBF-1 bean formulation (45.68%), while the lowest volume shrinkage was found in the MBF-2 formulation (0.86%). The apparent density of the chips ranged from 0.16 to 0.55 g/cm^3^, with significant differences (*p* < 0.05) among all formulations. The highest density was found in the MBF-1 beet formulation and the lowest in the MFC-1 chia and flaxseed formulation. Finally, weight loss (*p* < 0.05) was observed in the MFC-2 chia and flaxseed chips (51.03%), while the MBF-1 bean chips presented the highest loss with 29% (Table 3).

#### 3.2.3. Color Parameter of Chips

The lightness (*L**) of the chips ranged from 36.45 to 54.16 (Table 3), with significant differences observed between the formulations. Chips from the MFC-2 formulation (maize, chia, and linseed) had the highest lightness value of 54.16, indicating a light brown color, which enhances visual appeal [50]. These formulations could be the ones with the greatest sensory acceptance.

In contrast, beet chips had the lowest *L** values (MB-1: 31.66; MB-2: 30.25), resulting in a darker appearance compared to the other chips; this can be attributed to their higher betacyanin content, which contributes to darker tones [50]. Therefore, it is not perceived as an attractive feature for consumers.

Regarding the color parameter *a**, all chip formulations exhibited different shades of red, with significant differences (*p* < 0.05). Chips formulated with beans (MBF-1 and MBF-2) presented the highest *a** values, reflecting more intense red color tones; no significant differences were found between these two formulations, with an average of 5.83. The maize, chia, and flaxseed formulation (MCF-1: -0.57) showed a tendency towards green tones (Table 3).

In terms of the color parameter *b**, significant differences (*p* < 0.05) were also observed between all chips, each displaying different shades of yellow. The MCF-2 formulation achieved the highest *b** value at 18.89, indicating a more intense yellow hue. These findings underline the importance of ingredient selection in determining the sensory attributes of chips, which can significantly influence consumer preferences.

#### 3.2.4. Importance of Maize in the Diet

This section explores the importance of maize in the diet of the consumers who participated in the study. The results are illustrated in the word cloud generated from participants’ responses (Figure 2A). A remarkable 71.3% of respondents stated that they “always” participated in purchasing food for their family, while 26.5% indicated that they did so “sometimes”. This was because maize is a fundamental component of food systems, providing essential macronutrients and micronutrients, which are vital for health and nutrition [15].

In Honduran households, decision-making regarding the purchase of maize products is primarily the responsibility of the wife, as reported by 50.6% of respondents, while 24.1% indicated that it is a joint decision (wife and husband) (Figure 2B). Decision-making regarding maize products is predominantly influenced by household dynamics, emphasizing the role of maize as a staple food in Honduran households [15]. The most consumed maize products include fresh tortillas (34.81%), followed by chips (19.07%) and bread (16.30%) (Figure 2C). Regarding places of purchase, supermarkets are the most popular option (39.0%), followed by grocery stores (23.6%) and local markets (17.4%) (Figure 2D).

These results are in line with those reported by Escobar-López et al. [51] and Iuga et al. [52], who showed that women were the main decision-makers when purchasing food for the family, such as the purchase of tortillas. The preference for artisanal maize tortillas instead of industrial options highlights the importance of traditional food practices and their nutritional benefits [52].

Maize tortilla chip consumption patterns reveal important insights into consumer preferences and behaviors: 50% of respondents reported consuming maize tortilla chips once a week, while 18.6% consume them every two weeks, 17% consume them daily, and 10.1% consume them once a month (Figure 2E). The top three characteristics they consider when purchasing chips are flavor (18.8%), freshness (18.3%), and price (14.0%). In addition, they also mentioned the importance of color (12.3%) and texture (11.6%) (Figure 2F). The frequency of consuming maize tortilla chips weekly matches trends observed in salty snack consumption in the US, where 33% of adults reported eating salty snacks on a given day [53]. Key purchasing characteristics are flavor and freshness, which are central to the snack food market, which places emphasis on sensory attributes [54], and the price remains an important factor, reflecting broader trends in consumer behavior of cost influences snack food choices [53]. While the focus on flavor and freshness is predominant, some studies suggest that health considerations are increasingly influencing snack food choices, indicating a possible shift in consumer priorities [55].

#### 3.2.5. Sensory Evaluation by Consumers and Availability of Chips for Purchase

This section presents the results of the sensory evaluation and purchase willingness was carried out on 300 consumers. 57.10% were women and 42.90% were men. The age distribution showed that 58.95% of the respondents were between 31 and 60 years old, while 16% were over 61 years old. In addition, 14.20% of the participants were between 18 and 25 years old, and 10.49% were between 26 and 30 years old (Table 4). Regarding the level of education, Roughly half (49.38%) of the respondents had a university degree, while, 32.72% had completed secondary school, 9.26% had completed primary school, and 7.73% had completed postgraduate studies.

This is in line with the findings of other studies highlighting the influence of demographic variables on consumer behavior, particularly in food products and cosmetics. For example, women represent a larger segment of consumers and those with education tend to show greater engagement in purchasing decisions, while older consumers typically show greater interest in product labels and organic foods [56,57]. Furthermore, the educational level of respondents reveals that almost half (49.38%) have a college degree, which can be correlated with their purchasing preferences and behaviors [57]. Filiptsova et al. [58] mention that higher educational levels correlate with greater awareness and interest in product quality and safety.37.65% of respondents reported monthly income between HNL L 15,000 to 30,000 (approximately USD 610.42 to 1220.84 USD), while 30.86% received more than HNL 30,000 (more than USD 1220.84 USD), while 15.74% received a monthly income of less than HNL 15,000 (USD 610.42 USD);, the remaining 15.74% preferred not to answer. Finally, regarding family size, 25.93% of respondents answered that they had four members, followed by 21.91% with three members, and 10.49% with six members (Table 4). These results indicate that a significant part of the population has economic difficulties. Family size also has an influence, as 25.93% of respondents have four members. This income distribution can be compared to other countries in the Latin American region, highlighting the economic challenges faced by Honduran families. Despite these figures, other Central American countries may exhibit higher average incomes and lower poverty rates, suggesting that Honduras’ economic challenges are exacerbated by structural problems such as political instability and vulnerability to external shocks [59,60].

#### 3.2.6. Sensory Evaluation

We evaluated the following chip attributes: appearance, color, flavor, texture, and overall acceptance. Table 5 shows the average ratings assigned by consumers to each formulation. No significant differences (*p* > 0.05) were found between the sensory attributes evaluated, with an average liking value of 7.41 (“Moderately liked”) across all the chips evaluated (Table 5). These results can be attributed to the fact that the evaluated blends were pre-selected by a semi-trained panel for their favorable sensory characteristics (unpublished data). This is in line with the findings of several studies highlighting the importance of pre-selection of the formulation on consumer acceptability [61]. Perfilova et al. [50] and Reddy et al. [62] highlighted that the incorporation of non-traditional ingredients resulted in high sensory scores. This shows not only that innovative formulations increase consumer appeal but also that new flavors can have a significant impact on consumer acceptance. However, other studies also showed that consumer opinion can be positively influenced when informed about the nutritional characteristics of a product [7,63]. Lee and Lee [10] highlighted that sensory profiles in blind tests may differ from those of informed consumers, suggesting that perception plays a crucial role in acceptance.

The sensory evaluation also included a quick visual test, in which respondents selected which blend they found most healthy or nutritious; 47.2% of participants considered the maize chips with chia and flaxseed to be the most nutritious, while 31.8% chose the maize chips with beet, and 20.4% mentioned the maize and bean chips. This is consistent with findings that ordinary consumers often equate healthiness with nutritious natural ingredients. Importantly, consumer perceptions do not always reflect actual nutritional content, evidencing a potential discrepancy between nutritional education and marketing strategies [64].

The maize chips with chia and flaxseed were the most preferred due to taste, texture, and visual appearance, as indicated by the results of WAC and WSC (Table 2).

#### 3.2.7. Purchase Provision

Consumers’ willingness to purchase and pay a premium for the different chip mixes varied across the two purchase scenarios: one focused on daily use and the other on improving eating habits. As shown in Figure 3A, while participants were generally open to purchasing the chips in both scenarios, their willingness was notably higher in the scenario of improving eating habits, where approximately 95% indicated they were “totally sure” or “sure” of their purchase intention, consistent with findings showing a general willingness to pay a higher price for healthier options, which can range from 5.6% to 91.5% across studies [65]. Despite this positive inclination, only 40.2% of participants in both scenarios expressed willingness to pay more for chips compared to conventional options containing added artificial oils and preservatives, reflecting a common trend where price sensitivity limits premium purchases [66]. Meanwhile, 53.6% indicated that they would be willing to pay the same price as conventional chips, even after knowing the superior nutritional qualities of the baked product, which does not contain added oil, as mentioned by other authors [65].

In both scenarios, around 49.1% of participants were willing to pay 10% more compared to conventional chips. Furthermore, 16.7% in the daily use scenario and 20.3% in the scenario of improved eating habits stated that they would pay 20% more. In contrast, a small percentage of participants (36) expressed their willingness to pay 5% to 10% less than the price of conventional chips. This reluctance seems to be related to those consumers who showed a lower acceptance of the chips tasted during the sensory evaluation. This suggests that while health consciousness influences purchase intentions, economic factors remain a major barrier to actual premium pricing decisions [66]. In contrast, some studies indicate that demographic factors, such as age and education, may influence willingness to pay, with older adults and those with higher levels of education more likely to pay more for healthier foods [65,67].

## 4. Conclusions

This study highlights the potential of alternative flours derived from local ingredients, such as maize, beet, flaxseed, beans, and chia, to improve the nutritional composition of traditional food products, in this case, tortilla chips. Flaxseed flour had a high protein content, while chia flour stood out for its high lipid and fiber content. In addition, both bean and chia flours are rich in essential minerals such as iron and zinc. Fiber, particularly insoluble fiber in bean flour and soluble fiber in flaxseed flour, reinforces their potential to formulate functional foods focused on digestive health and cholesterol regulation. The high-water retention and gel-forming capacity of chia flour, together with the oil absorption capacity of flaxseed flour, improved texture, structural stability, and sensory characteristics. Consumers demonstrated a general preference for corn–chia–flaxseed blends (47.2%), indicating a positive receptivity towards more nutritious products; 50% of participants indicated that they consume tortilla chips weekly, prioritizing taste, freshness, and price, but more than 40% showed willingness to pay a premium price for baked and more nutritious options. These results suggest that the use of alternative flours can not only enrich the local diet, but also promote healthier eating habits in the population. Furthermore, the positive response from consumers highlights a market opportunity for small and medium-sized enterprises (SMEs), who could take advantage of this trend to increase awareness about nutrition and public health in Honduras, which in turn could have a positive impact on the country’s collective health.

## Figures and Tables

**Figure 1 foods-14-00864-f001:**
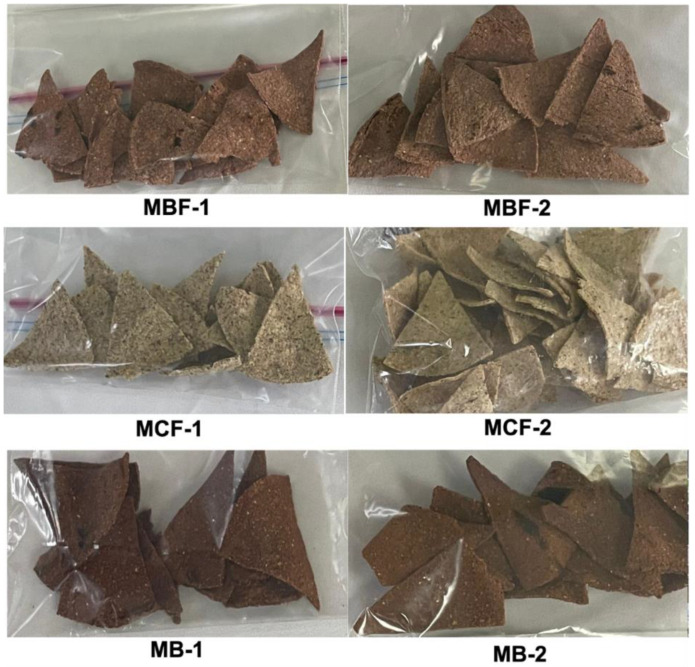
Photographs of the chips evaluated.

**Figure 2 foods-14-00864-f002:**
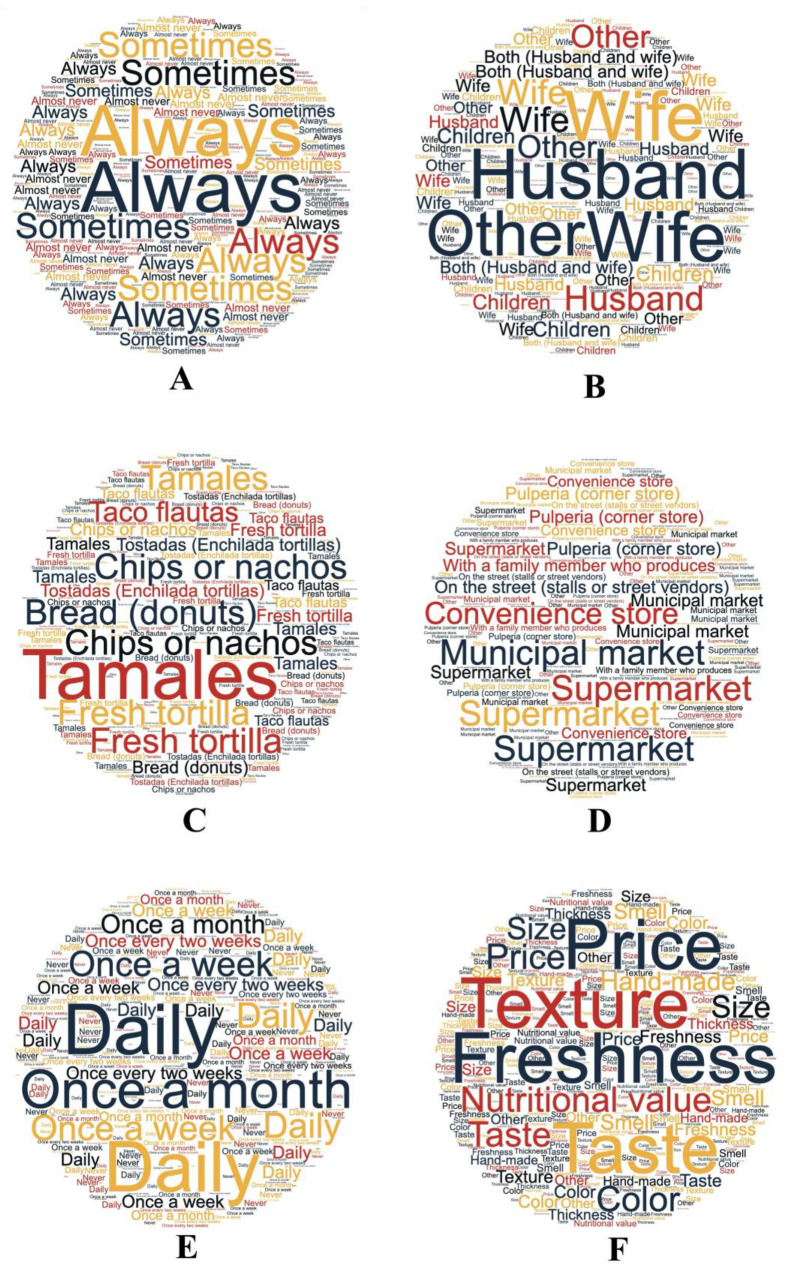
Importance of maize in the diet: (**A**) Participation in food purchasing, (**B**) Food purchasing decision. (**C**) Principal maize products in the family diet, (**D**) Places that sell maize products, (**E**) Chip consumption frequency, (**F**) Principal attributes in the purchasing decision.

**Figure 3 foods-14-00864-f003:**
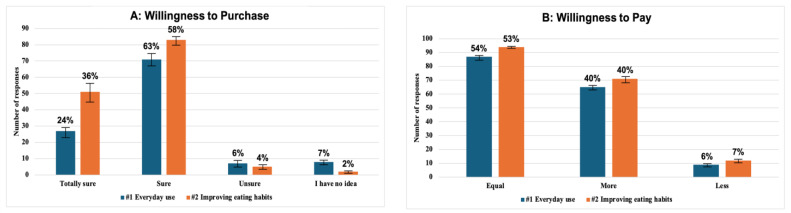
Consumer willingness to purchase: (**A**) Willingness to purchase, (**B**) Willingness to pay.

**Table 1 foods-14-00864-t001:** Tortilla chip formulation (g/100 g).

No.	Ingredients	Tortilla Chips					
		MBF-1	MBF-2	MCF-1	MCF-2	MB-1	MB-2
1	Maize flour (g)	55.00	73.66	76.67	90.00	85.00	90.00
2	Chia flour (g)	--	--	11.67	5.00	--	--
3	Flaxseed flour (g)	--	--	11.67	5.00	--	--
4	Bean flour (g)	45.00	26.34	--	--	--	--
5	Beetroot flour (g)	--	--	--	--	15.00	10.00

MBF: maize and bean chips; MCF: maize, chia, and flaxseed chips; MB: maize and beet chips; 1: formulation 1; 2: formulation 2.

**Table 2 foods-14-00864-t002:** Chemical composition, color, and techno-functional properties of flours.

Property	Maize Flour	Bean Flour	Chia Flour	Flaxseed Flour	Beet Flour
Chemical composition					
Moisture (g/100 g)	10.37 ± 0.21	9.39 ± 0.19	7.35 ± 0.15	6.77 ± 0.14	5.38 ± 0.11
Ash (g/100 g)	1.31 ± 0.03	3.92 ± 0.08	4.05 ± 0.08	2.76 ± 0.06	7.07 ± 0.14
Al (mg/kg)	2.89 ± 0.26	6.70 ± 0.07	4.35 ± 0.09	17.39 ± 0.32	13.36 ± 0.07
Cd (mg/kg)	0.04 ± 0.06	0.09 ± 0.01	0.10 ± 0.03	0.53 ± 0.07	0.14 ± 0.00
Cr (mg/kg)	0.22 ± 0.02	0.42 ± 0.03	0.31 ± 0.09	0.67 ± 0.01	0.58 ± 0.02
Cu (mg/kg)	2.05 ± 0.37	7.16 ± 0.10	6.07 ± 0.28	14.82 ± 0.05	16.37 ± 0.13
Fe (mg/kg)	17.75 ± 0.34	73.36 ± 1.18	27.59 ± 0.00	69.53 ± 0.53	58.15 ± 0.38
Mn (mg/kg)	5.37 ± 0.37	20.90 ± 0.16	193.41 ± 1.34	24.74 ± 0.38	76.46 ± 0.70
Ti (mg/kg)	0.65 ± 0.14	0.22 ± 0.01	0.22 ± 0.27	1.90 ± 0.22	0.16 ± 0.17
Zn (mg/kg)	21.25 ± 0.31	208.36 ± 2.61	80.85 ± 0.10	42.99 ± 1.28	48.65 ± 0.19
Ca (mg/kg)	636.49 ± 0.69	1930.19 ± 1.00	1175.19 ± 60.12	1744.27 ± 62.16	6443.17 ± 3.19
K (mg/kg)	3352.87 ± 26.61	12,888.95 ± 93.24	23,176.89 ± 30.21	7305.96 ± 76.52	6784.84 ± 72.74
P (mg/kg)	2654.61 ± 54.54	4738.06 ± 66.86	2303.20 ± 85.86	5365.42 ± 80.23	6651.24 ± 4.31
Fat (g/100 g)	4.23 ± 0.08	1.13 ± 0.02	32.25 ± 0.64	27.80 ± 0.56	0.14 ± 0.00
Protein (g/100 g)	8.17 ± 0.00	23.57 ± 0.30	31.84 ± 0.15	40.03 ± 0.12	12.95 ± 0.11
Lys (g/100 g)	0.241 ± 0.01	0.659 ± 0.01	1.132 ± 0.14	0.847 ± 0.01	0.112 ± 0.01
Trp (g/100 g)	0.026 ± 0.00	0.042 ± 0.00	0.111 ± 0.01	0.606 ± 0.02	0.422 ± 0.01
Total dietary fiber (g/100 g)	12.74 ± 0.25	39.74 ± 0.79	38.51 ± 0.77	27.42 ± 0.55	7.03 ± 0.14
Insoluble dietary fiber (g/100 g)	12.27 ± 0.25	35.72 ± 0.71	32.99 ± 0.66	13.61 ± 0.27	5.49 ± 0.11
Soluble dietary fiber (g/100 g)	0.47 ± 0.01	4.02 ± 0.08	5.52 ± 0.11	13.81 ± 0.28	1.54 ± 0.03
Color					
*L**	33.38 ± 0.47	15.12 ± 0.21	14.60 ± 0.20	10.32 ± 0.14	9.44 ± 0.13
*a**	−6.77 ± 0.09	9.56 ± 0.13	6.02 ± 0.08	2.12 ± 0.03	8.94 ± 0.12
*b**	3.05 ± 0.04	5.76 ± 0.08	8.36 ± 0.12	3.19 ± 0.04	5.24 ± 0.07
Techno-functional properties					
WAC (g H_2_O/g sample)	3.75 ± 0.35	3.80 ± 0.14	9.15 ± 0.35	3.25 ± 0.21	4.15 ± 0.21
WSC (%)	15.19 ± 0.74	17.18 ± 0.53	69.84 ± 12.35	13.85 ± 0.62	20.42 ± 0.59
OAC (g oil/g sample)	2.25 ± 0.07	2.55 ± 0.21	2.30 ± 0.00	2.65 ± 0.07	2.45 ± 0.07

Means ± SD of three determinations; N.D: not detected; WAC: water absorption capacity; WSC: water solubility capacity; OAC: oil absorption capacity.

**Table 3 foods-14-00864-t003:** Physical and chemical properties of different tortilla chip formulas.

Property	MBF-1	MBF-2	MCF-1	MCF-2	MB-1	MB-2
Chemical composition						
Moisture (g/100 g)	8.17 ± 0.16a	9.14 ± 0.18b	9.43 ± 0.19bc	9.25 ± 0.18bc	9.57 ± 0.19cd	9.84 ± 0.20d
Ash (g/100 g)	2.65 ± 0.05f	1.54 ± 0.03b	1.90 ± 0.04d	1.20 ± 0.02a	2.04 ± 0.04e	1.76 ± 0.04c
Al (mg/kg)	5.02 ± 0.01d	3.18 ± 0.09a	5.29 ± 0.33de	4.51 ± 0.44c	5.73 ± 0.23e	3.71 ± 0.25b
Cd (mg/kg)	0.04 ±0.04a	0.04 ± 0.05a	0.11 ± 0.16a	0.14 ± 0.06a	0.07 ± 0.01a	0.07 ± 0.02a
Cr (mg/kg)	0.30 ± 0.01ab	0.26 ± 0.02ab	0.41 ± 0.04c	0.31 ± 0.03b	0.29 ± 0.03ab	0.25 ± 0.04a
Cu (mg/kg)	4.55 ± 0.36c	3.04 ± 0.19b	4.19 ± 0.15c	3.44 ± 0.54b	2.38 ± 0.05a	2.08 ± 0.02a
Fe (mg/kg)	44.56 ± 0.58f	32.53 ± 0.22e	25.07 ± 0.66d	20.60 ± 0.49c	18.88 ± 0.27b	17.87 ± 0.20a
Mn (mg/kg)	12.77 ±0.06c	9.20 ± 0.27a	13.02 ± 0.05c	10.50 ± 0.11b	36.96 ± 0.88e	28.69 ± 0.36d
Ti (mg/kg)	0.30 ± 0.33ab	0.15 ± 0.05a	0.26 ± 0.21ab	0.32 ± 0.03ab	0.26 ± 0.12ab	0.56 ± 0.18b
Zn (mg/kg)	95.14 ± 0.07e	62.59 ± 0.41d	21.72 ± 0.37a	21.11 ± 0.19a	30.26 ± 0.63c	27.89 ± 0.12b
Ca (mg/kg)	1272.66 ± 7.95f	1076.74 ± 2.06d	1217.91 ± 18.07e	1018.08 ± 9.45c	761.77 ± 12.55b	702.22 ± 16.13a
K (mg/kg)	8539.05 ± 67.27e	6498.44 ± 44.43c	3803.74 ± 54.33b	3503.41 ± 61.93a	7427.03 ± 49.29d	6409.55 ± 17.97c
P (mg/kg)	3745.47 ± 61.92e	3150.18 ± 3.67d	2948.75 ± 95.23c	2811.73 ± 90.81b	2549.29 ± 27.59a	2557.28 ± 49.81a
Fat (g/100 g)	2.91 ± 0.06a	3.31 ± 0.07b	11.07 ± 0.22e	6.71 ± 0.13d	3.43 ± 0.07b	3.80 ± 0.08c
Protein (g/100 g)	14.78 ± 0.16e	12.17 ± 0.04d	11.48 ± 0.10c	10.18 ± 0.12b	9.15 ± 0.08a	9.12 ± 0.05a
Lys (g/100 g)	0.503 ± 0.02c	0.458 ± 0.01c	0.481 ± 0.04c	0.476 ± 0.02c	0.310 ± 0.06a	0.376 ± 0.00b
Trp (g/100 g)	0.036 ± 0.00a	0.034 ± 0.00a	0.076 ± 0.01b	0.045 ± 0.02a	0.038 ± 0.01a	0.032 ± 0.00a
Total dietary fiber (g/100 g)	25.86 ± 0.52e	16.72 ± 0.33b	24.33 ± 0.49d	21.61 ± 0.43c	10.81 ± 0.22a	10.53 ± 0.21a
Insoluble dietary fiber (g/100 g)	25.09 ± 0.50e	16.54 ± 0.23b	19.69 ± 0.39d	17.49 ± 0.35c	10.73 ± 0.21a	10.51 ± 0.20a
Soluble dietary fiber (g/100 g)	0.77 ± 0.02c	0.18 ± 0.01b	4.63 ± 0.09e	4.12 ± 0.08d	0.08 ± 0.00a	0.03 ± 0.00a
Carbohytrates (g/100 g)	45.63 ± 0.95b	57.12 ± 0.65d	41.79 ± 1.04a	51.05 ± 0.88c	65.00 ± 0.60e	64.95 ± 0.18e
Physical properties						
Weight (g)	3.55 ± 0.06c	3.70 ± 0.06d	3.28 ± 0.05b	2.98 ± 0.05a	3.32 ± 0.05b	3.48 ± 0.06c
Diameter (mm)	26.69 ± 0.43a	27.57 ± 0.44b	29.51 ± 0.47c	29.26 ± 0.47c	26.28 ± 0.42a	27.74 ± 0.44b
Thickness (mm)	1.94 ± 0.03f	1.81 ± 0.03e	1.58 ± 0.03d	1.14 ± 0.02a	1.27 ± 0.02b	1.35 ± 0.02c
Degree of contracting (%)	32.24 ± 0.52f	15.94 ± 0.26e	12.05 ± 0.19b	10.19 ± 0.16a	14.03 ± 0.22c	14.57 ± 0.23d
Degree of volume contraction (%)	45.68 ± 0.73e	0.86 ± 0.01a	12.87 ± 0.21c	35.84 ± 0.57d	10.27 ± 0.16b	9.73 ± 0.16b
Apparent density (g/cm^3^)	0.28 ± 0.00c	0.30 ± 0.00d	0.20 ± 0.00b	0.16 ± 0.00a	0.55 ± 0.01f	0.47 ± 0.01e
Weight loss (%)	29.00 ± 0.46a	43.08 ± 0.69b	48.83 ± 0.78c	51.03 ± 0.82d	43.97 ± 0.70b	44.18 ± 0.71b
Color						
*L**	42.11 ± 0.58e	40.84 ± 0.06d	36.45 ± 0.67c	54.16 ± 0.43f	31.66 ± 0.07b	30.25 ± 0.51a
*a**	5.52 ± 0.16de	6.14 ± 0.59e	−0.57 ± 0.07a	0.49 ± 0.66b	4.19 ± 0.09c	5.26 ± 0.07d
*b**	9.23 ± 0.36b	10.39 ± 0.75c	11.89 ± 0.61d	18.93 ± 0.65e	2.86 ± 0.06a	11.59 ± 0.65d

MBF: maize and bean chips; MCF: maize, chia, and flaxseed chips; MB: maize and beet chips; 1: formulation 1; 2: formulation 2. Means ± SD of three determinations showing the same letters in the same row are not significantly different (Fisher *p* < 0.05).

**Table 4 foods-14-00864-t004:** Sociodemographic profile of respondents.

Sociodemographic Profile	Classification	Frequency	Percent (%)
Gender	Male	139	42.90
	Female	185	57.10
	Non-binary	0	0
Age range	18 to 25 years	46	14.20
	26 to 30 years	34	10.49
	31 to 60 years	191	58.95
	Over 61 years	53	16.36
Education level	No schooling	1	0.31
	Elementary school	30	9.26
	Highschool	106	32.72
	Bachelor’s degree	160	49.38
	Postgraduate	27	8.34
Household composition	One member	12	3.70
	Two members	24	7.41
	Three members	71	21.91
	Four members	84	25.93
	Five members	71	21.91
	More than six members	62	18.82
Household income range	Below USD 600	51	15.74
	USD 600 to USD 1200	122	37.65
	Above USD 1200	100	30.86
	No response	51	15.74

**Table 5 foods-14-00864-t005:** Sensory scores of different tortilla chip formulas.

Attribute	Tortilla Chips					
	MBF-1	MBF-2	MCF-1	MCF-2	MB-1	MB-2
Appearance	7.58 ± 1.37a	7.35 ± 1.44a	7.75 ± 1.57a	7.47 ± 1.48a	7.19 ± 1.58a	7.13 ± 1.53a
Color	7.72 ± 1.50a	7.10 ±1.58a	7.72 ± 1.65a	7.51 ±1.50a	7.09 ± 1.61a	7.27 ± 1.49a
Flavor	7.63 ± 1.68a	6.98 ± 1.74a	7.80 ± 1.76a	7.58 ±1.70a	7.28 ± 1.80a	7.12 ± 1.72a
Texture	7.83 ± 1.41a	7.53 ± 1.63a	7.91 ± 1.48a	7.64 ± 1.66a	7.52 ± 1.71a	7.47 ± 1.69a
Overall acceptability	7.69 ± 1.49a	7.24 ± 1.60a	7.80 ± 1.62a	7.55 ± 1.59a	7.27 ± 1.68a	7.25 ± 1.61a

MBF: maize and bean chips; MCF: maize, chia, and flaxseed chips; MB: maize and beet chips; 1: formulation 1; 2: formulation 2. Means ± SD showing the same letters in the same row are not significantly different (Fisher *p* < 0.05).

## Data Availability

The original contributions presented in the study are included in the article; further inquiries can be directed to the corresponding author.

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
