# Peer review of "Nutritional Benefits and Consumer Acceptance of Maize Chips Combined with Alternative Flours"

_foods, 2025, doi:10.3390/foods14050864_

Round 1

Reviewer 1 Report

Comments and Suggestions for Authors

Line 27- please expand the abbreviation of SME

Line 42-43- please remember that beetroot does not contain anthocyanins. The natural pigments of beetroot are betalains, especially betanin. Please read and cite literature: https://doi.org/10.3390/molecules28052018 ; https://doi.org/10.3390/molecules29204803

The introduction should be expanded with a paragraph about recent studies concerning the staple food fortification with different ingredients and its effectiveness. The authors focus mainly on describing the trend itself, consumer acceptability and the subject of supplementation maize chips.

Line 71- please expand the abbreviation of SME

The purpose and scope of the study are clearly defined. The authors point out the benefits of studying the issue presented.

Line 86-89- please provide range of particle size like you did for other flours

Preparation of flours- please provide the model name of oven and conditions for dehydration of each flour

Physicochemical characterization of raw materials and chips – please provide the full methodology description for all determinations mentioned here. Separate the elements from the ash, and the two amino acids from the protein by removing the brackets. I don't know how you determined anthocyanins, but there are none in beet. I ask for clarification.

Process of making tortilla chips – provide the names of machines used for making process

Line 148- please provide mentioned data as supplemented file

Line 170- Table 2? Please check table 2 is the composition of flours not survey. Please provide survey questions in supplemented file.

Line 204-208, 292-294 – beetroot does not contain anthocyanins, it contain betalains (betacyanins and vulgaxanthins)

Please modify the names for subsection 3.2.5 and 3.2.6 because they are nearly the same

The discussion of the chemical composition, physical and sensory evaluation of the flour and chips are not objectionable (except for anthocyanins) and are supported by the literature.

Conclusions

line 481-484 – please delete this sentence because this is a repetition of information already provided in introduction and tables. It is obvious that this ingredients have various composition and functional properties.  These are generalities.

Line 485-486- “….while beet stands out for its richness in anthocyanins.” – beetroot does not contain anthocyanins!

The conclusions are very sparse for the amount of analysis and research done by the authors. They should be expanded, first of all, on whether it was effective the increasing of the nutritional value of the different variants of chips.

Author Response

Comments attended are given below:

All corrections are highlighted in yellow on the manuscript.

Comments to the Author

Reviewer #1

R#1 Line 27- please expand the abbreviation of SME

Answer: Added:…. Small and Medium-sized Enterprises

R#1 Line 42-43- please remember that beetroot does not contain anthocyanins. The natural pigments of beetroot are betalains, especially betanin. Please read and cite literature: https://doi.org/10.3390/molecules28052018 ; https://doi.org/10.3390/molecules29204803

Answer: Thanks for the comment, It was decided to eliminate that determination.

R#1 The introduction should be expanded with a paragraph about recent studies concerning the staple food fortification with different ingredients and its effectiveness. The authors focus mainly on describing the trend itself, consumer acceptability and the subject of supplementation maize chips.

Answer: Thanks for the comment.

Paragraph added: In this sense, Vasisht et al. [5] evaluated the effect of pomegranate peel on the physicochemical and antioxidant properties of tortilla chips prepared from germinated maize and mung bean flour, as a healthier snack alternative. They recommend 5% concentration of peel fried at 180 ºC, preserving the sensory attributes and nutrients. On the other hand, Buzgau et al. [6] evaluated the chemical composition and quality characteristics of tortilla chips made from maize flour with chickpea flour and red lentil flour. This study demonstrated that the incorporation of chickpea flour in maize flour chips improves the nutritional qualities of tortilla chips. Zuñiga-Martínez et al. [7] studied the effect of adding avocado paste (AP) to maize chips on their nutritional profile and sensory acceptability. They found that 2% AP was the best concentration to improve the nutritional properties of maize chips without compromising their sensory acceptability.

Reference added: Vasisht, R.; Yadav, R.B.; Yadav, B. S. Effect of pomegranate peel on physicochemical and antioxidant properties of tortilla chips prepared from germinated corn and mung bean flour. Future Foods, 2024, 9, 100363.

R#1 Line 71- please expand the abbreviation of SME

Answer: Added:…. Small and Medium-sized Enterprises

R#1 The purpose and scope of the study are clearly defined. The authors point out the benefits of studying the issue presented.

Answer: Thanks for the comment.

R#1 Line 86-89- please provide range of particle size like you did for other flours

Answer: Added:….. (Until a particle size of approximately 0.5 mm)

R#1 Preparation of flours- please provide the model name of oven and conditions for dehydration of each flour

Answer: The brand, model and country of the ovens and mills used were added.

R#1 Physicochemical characterization of raw materials and chips – please provide the full methodology description for all determinations mentioned here. Separate the elements from the ash, and the two amino acids from the protein by removing the brackets. I don't know how you determined anthocyanins, but there are none in beet. I ask for clarification.

Answer: The methodology and the brands and models of the equipment used were added.

R#1 Process of making tortilla chips – provide the names of machines used for making process

Answer: Thanks for your comment, the brands and models of the equipment used have been added.

R#1 Line 148- please provide mentioned data as supplemented file

Answer: Thank you for the comment. The mentioned data is not provided as it will be part of another article that is being written.

R#1 Line 170- Table 2? Please check table 2 is the composition of flours not survey. Please provide survey questions in supplemented file.

Answer: Thanks for your comment, Deleted: (Table 2).

R#1 Line 204-208, 292-294 – beetroot does not contain anthocyanins, it contain betalains (betacyanins and vulgaxanthins)

Answer: Thanks for your comment. It was decided to eliminate that determination.

R#1 Please modify the names for subsection 3.2.5 and 3.2.6 because they are nearly the same

Answer: The names for subsections 3.2.5 and 3.2.6 are completely different. 3.2.5 Sensory Evaluation and 3.2.6. Purchase provision.

R#1 The discussion of the chemical composition, physical and sensory evaluation of the flour and chips are not objectionable (except for anthocyanins) and are supported by the literature.

Answer: Thanks for your comment. It was decided to eliminate that determination.

R#1 Conclusions

line 481-484 – please delete this sentence because this is a repetition of information already provided in introduction and tables. It is obvious that this ingredients have various composition and functional properties. These are generalities.

Answer: Thanks for your comment. Deleted: Lines 481-484.

R#1 Line 485-486- “….while beet stands out for its richness in anthocyanins.” – beetroot does not contain anthocyanins!

Answer: Thanks for your comment. It was decided to eliminate that determination.

R#1 The conclusions are very sparse for the amount of analysis and research done by the authors. They should be expanded, first of all, on whether it was effective the increasing of the nutritional value of the different variants of chips.

Answer: The conclusion was written again.

The study highlights the potential of alternative flours derived from local ingredients, such as maize, beet, flaxseed, beans and chia, to improve the nutritional composition of traditional food products, in this case, tortilla chips. Flaxseed flour had a high protein content, while chia flour stood out for its high lipid and fiber content. In addition, both bean and chia flour are rich in essential minerals such as iron and zinc. Fiber, particularly insoluble in bean flour and soluble in flaxseed flour, reinforces their potential to formulate functional foods focused on digestive health and cholesterol regulation. The high-water retention and gel-forming capacity of chia flour, together with the oil absorption capacity of flaxseed flour, improved texture, structural stability and sensory characteristics. Consumers demonstrated a general preference for corn-chia-flaxseed blends (47.2%), indicating a positive receptivity towards more nutritious products. 50% of participants indicated that they consume tortilla chips weekly, prioritizing taste, freshness, and price, but more than 40% showed willingness to pay a premium price for baked and more nutritious options. These results suggest that the use of alternative flours can not only enrich the local diet, but also promote healthier eating habits in the population. Furthermore, the positive response from consumers highlights a market opportunity for small and medium-sized enterprises (SMEs), who could take advantage of this trend to increase awareness about nutrition and public health in Honduras, which in turn could have a positive impact on the country's collective health.

Reviewer 2 Report

Comments and Suggestions for Authors

The manuscript investigates the nutritional benefits and consumer acceptance of maize chips combined with alternative flours, a topic of growing interest in both food science and public health. The study design is commendable, incorporating chemical, functional, and sensory analyses. However, the study demonstrates certain weaknesses in its scientific depth, methodological rigor, and overall organization. The following issues must be addressed for the manuscript to be considered for publication.

1. While properties like water absorption capacity (WAC), water solubility capacity (WSC), and oil absorption capacity (OAC) are measured, their implications for the dough structure, sensory attributes, and product stability are not sufficiently explored.

2. While the sensory evaluation includes a large sample size (300 participants), it lacks detailed analysis of what drives consumer preferences. For instance, why were maize-chia-flaxseed chips most preferred? Was it due to flavor, texture, or appearance? The analysis is descriptive rather than inferential.

3. Weak Integration of Results with Existing Literature: While the article presents comprehensive data, it does not sufficiently compare the results with findings from similar studies. For instance, nutritional advantages of bean and chia flours are discussed, but their performance compared to similar formulations in other studies is not addressed.

4. The article does not adequately justify why certain methods and ingredient proportions were chosen. For example, why were specific proportions of alternative flours selected? Were the ratios based on optimizing sensory qualities, functional properties, or nutritional goals?

5. While the study assesses water absorption capacity (WAC), water solubility capacity (WSC), and oil absorption capacity (OAC), the methods are described briefly without detailing the protocols or equipment used.

6. The manuscript presents the chemical composition, functional properties, and sensory data as separate results without integrating them into a cohesive discussion. Author should link the findings across these sections.

7. Please ensure all graphs include p-values or error bars to validate comparisons between formulations (Table 2).

Comments on the Quality of English Language

the english writing is fine

Author Response

Comments attended are given below:

All corrections are highlighted in yellow on the manuscript.

Comments to the Author

Reviewer #2

R#2 The manuscript investigates the nutritional benefits and consumer acceptance of maize chips combined with alternative flours, a topic of growing interest in both food science and public health. The study design is commendable, incorporating chemical, functional, and sensory analyses. However, the study demonstrates certain weaknesses in its scientific depth, methodological rigor, and overall organization. The following issues must be addressed for the manuscript to be considered for publication.

R#2 1. While properties like water absorption capacity (WAC), water solubility capacity (WSC), and oil absorption capacity (OAC) are measured, their implications for the dough structure, sensory attributes, and product stability are not sufficiently explored.

Answer: Thanks for the comment. The following paragraphs were added:

……… Chia flour contributes to a cohesive dough structure in food products, improving the integrity of chips during processing and storage. The gel-forming ability of chia flour can help maintain the structural integrity of chips, preventing them from breaking and ensuring a desirable crunch [3]. Khan et al. [31] mention that the incorporation of chia flour improved sensory attributes such as texture and mouthfeel, making chips more appealing to consumers.

……. This potential is supported by its rich nutritional profile and functional properties, making it suitable for fat-based applications. The high OAC content improves the sensory at-tributes of food products [32]. Hussain et al. [33] mention that roasting flaxseed meal improves its water absorption capacity and reduces antinutritional factors, making it more suitable for various food applications, and would favor the preparation of chips.

R#2 2. While the sensory evaluation includes a large sample size (300 participants), it lacks detailed analysis of what drives consumer preferences. For instance, why were maize-chia-flaxseed chips most preferred? Was it due to flavor, texture, or appearance? The analysis is descriptive rather than inferential.

Answer: Thanks for the comment. Added:

The maize chips with chia and flaxseed were the most preferred due to taste, texture and visual appearance, as indicated by the results of WAC and WSC (Table 2).

R#2 3. Weak Integration of Results with Existing Literature: While the article presents comprehensive data, it does not sufficiently compare the results with findings from similar studies. For instance, nutritional advantages of bean and chia flours are discussed, but their performance compared to similar formulations in other studies is not addressed.

Answer: Thanks for the comment. Added:

…….However, this ash content is lower than that reported by Coorey et al. [3] in chips made from a mixture of maize flour and chia flour (5.89-7.33 g/100 g), and higher than that reported in chips made from a mixture of maize and avocado paste (1.52-1.82 g/100 g) [7].

…… For example: In that reported in chips of mixtures of maize and avocado paste (2.08-3.62 g/100 g) [7] and that reported in corn chips with grape seed extracts (5.3-5.7 g/100 g) [39] and much lower than that reported in chips of mixtures of pomegranate peel, germinated maize and mung bean flour fried by immersion in oil (21.33-30.00 g/100 g) [5]. Therefore, the low lipid content of the chips developed in this research is evident compared to chips made in the traditional way (Fried by immersion in oil).

….. The results in the maize and bean formulations coincide with Hurtado et al. [40] 13.3-16.6 g/100 g and Ochoa-Martínez et al. [17] 13.00-13-28 g/100 g, who reported a high protein content in mixtures of these in-gredients.

…… The fiber content found in chips with maize flour, chia flour and flaxseed flour is higher than that reported in chips made from a mixture of maize and avocado paste (2.72-5.29 g/100 g) [7], chips made from a mixture of maize flour and chia flour (5.89-7.33 g/100 g) [3] and in chips made from a mixture of maize flour and bean flour (11.53-19.21 g/100 g) [38].

R#2 4. The article does not adequately justify why certain methods and ingredient proportions were chosen. For example, why were specific proportions of alternative flours selected? Were the ratios based on optimizing sensory qualities, functional properties, or nutritional goals?

Answer: The selection of the flour proportions was based on a preliminary study. The following experimental designs were used: D-optimal mixtures for the HM/HF mixtures (maize flour/bean flour, respectively) and for the HM/HR mixtures (maize flour/beet flour, respectively), with 10 different mixtures for each of the formulations. As well as a Simplex Lattice Experimental Design for the HF/HL/HC mixtures (maize flour/linseed flour/chia flour, respectively), 14 different mixtures.

Selecting the best proportions according to the preferences of a semi-trained sensory panel, two chips of the mixtures were selected per formulation, based on the best sensory attributes (flavor, color, appearance, texture and general acceptance). The mentioned data is not provided as it will be part of another article that is being written.

The paragraph was modified:

Chips were prepared using different proportions of each blend: maize/bean (MBF), maize/chia/flaxseed (MCF) and maize/beet (MB). Two formulations for each blend were selected according to the preferences of a semi-trained sensory panel, based on the best sensory attributes (flavor, color, appearance, texture and overall acceptance, unpublished data). The proportions of each of the flours and formulations are shown in Table 1 and their appearance in Figure 1.

R#2 5. While the study assesses water absorption capacity (WAC), water solubility capacity (WSC), and oil absorption capacity (OAC), the methods are described briefly without detailing the protocols or equipment used.

Answer: Thanks for the comment. The methodology and specifications of the equipment used were added, as well as the entire M&M section.

In the raw materials, the following were evaluated: Water absorption capacity (WAC) and Water solubility capacity (WSA): 1 g of sample was weighed in centrifuge tubes and 10 mL of distilled water was added, shaken for 30 s in a vortex (Vortex-2 Genie, Model G-560, Scientific Industries, INC, Bohemia, N.Y. USA) and centrifuged for 3500 rpm/15 min (Rotina 380 R, Hettich, Tuttlingen, Germany). The results were expressed as grams of water retained per gram of sample and for WSA as a percentage. Oil absorption capacity (OAC): 1 g of sample was weighed and 10 mL of maize oil was added and shaken for 30 s in a vortex mixer (Vortex-2 Genie, Model G-560, Scientific Industries, INC, Bo-hemia, N.Y. USA) and centrifuged at 3500 rpm for 15 min (Rotina 380 R, Hettich, Tut-tlingen, Germany). The results were expressed as grams of oil retained per gram of sample [12].

R#2 6. The manuscript presents the chemical composition, functional properties, and sensory data as separate results without integrating them into a cohesive discussion. Author should link the findings across these sections.

Answer: Thanks for the comment. Some lines have been added throughout the manuscript to further integrate the results and discussion.

R#2 7. Please ensure all graphs include p-values or error bars to validate comparisons between formulations (Table 2).

Answer: Thanks for the comment. Error bars have been added to Figure 3. Table 2 includes the standard deviation.

Round 2

Reviewer 1 Report

Comments and Suggestions for Authors

The authors have made all the recommended corrections and additions. They significantly improved the chapter with conclusions and work methodology. The authors added current knowledge in the topics covered in the introduction. Most importantly, the controversial methodology and results from the analysis of anthocyanins, which should not be present in the determined raw materials, were removed from the work. Apparently, the authors were not sure what they were determining. It's a pity they didn't comment in the response to reviewers on the original inclusion of these data in the manuscript.

Reviewer 2 Report

Comments and Suggestions for Authors

the raised comments were properly addressed, and the manuscript can be considered for publication

Comments on the Quality of English Language

English is fine